# The Significance of 'the Person' in Addiction

**Pádraic Mark Hurley**

Department of Applied Arts, Waterford Institute of Technology, X91 KOEK Waterford, Ireland;
padraic.hurley@wit.ie

**Abstract:** Van Gordon et al. outline the classification of their Ontological Addiction Theory (OAT), including its aetiology and treatment. In this review article I will from an appreciative perspective question some of its fundamental assumptions by presenting an alternative view on the ontology of 'the person', as distinct from its presently assumed conventional conflation with a contracted separate egoic self. I will propose this view as structurally and ethically significant for the 'embodied' experience of a reconstructed "dynamic and non-dual self", as cultivated in their treatment. Rather than this reconstructed self simply being socially desirable for functional purposes, I will underscore the meaning-generative case for ontological status, in the absence of which, a pervasive 'sense of lack' is evident, with all attendant individual, psychological, social, ecological and ethical implications. This article brings a developmental psychology perspective to bear in appreciating 'personhood' as an emergent, progressively realised and is thus similarly aligned with the intent of OAT in overcoming egoic addictive suffering. This mapping of the territory however populates a blind spot in OAT's diagnosis by affirming unique personhood, a quality of 'integrative presence', meaningfully understood as a psycho-spiritual ontological reality. It offers, as with OAT's stated intent, the merit of avoiding attendant mental health and developmental pitfalls, which can beset what we may discern as an implicit transcendental reductionist assumption operative in OAT, where 'the many' are reduced to 'the One' and there are, it is assumed, no real many. This framing is resonant with the lived experience of healthy 'individuation', a process distinct from the problematic phenomenon of 'individualism', evidenced by the empirical data on post-conventional human development, which potentially provides diagnostic markers for any optimal treatment discernment. It is also attuned to what many recognise as a contemporary Fourth Turning in Buddhism, in its conscious evolutionary recognition of the emergence in non-dual states of a 'unique personal perspective', and/or a relative individuation within the whole. This differentiation has formerly been interpreted through an 'impersonal' lens as an egoic holdover, and potentially inhibits ethical action in the world, as distinct from the ethical import and potential fruits stemming from the ontological affirmation of the person.

**Keywords:** ego; unique personhood; ontology; epistemology; contemplative traditions; Western Enlightenment; developmental psychology; transcendental reductionism; Fourth Turning in Buddhism

## 1. Introduction

Van Gordon et al. (2016), in laying out OAT, chart a path from (i) becoming aware of the imputed self, (ii) deconstructing the imputed self, and (iii) reconstructing a dynamic and non-dual self in order to overcome the ontological addiction, described as a "maladaptive condition whereby an individual is addicted to the belief that they inherently exist" (ibid, p. 1). And indeed, overcoming egoic addictive suffering is at the core of contemplative traditions worldwide, whilst subtle significant distinctions remain as to respective 'self-systems'. Set within a conscious evolutionary frame, some of the contemporary mapping of this territory, only relatively recent in its articulation, holds profound implications for our way of being and becoming in the world. Our conceptions of 'reality' and 'illusion', what we value and strive for, our sense of purpose and the very meaning of our lives are all at play. We are therefore exhorted to *choose* our 'view' wisely.

## 2. Discussion

The evolutionary nature of our cosmos, as we presently understand it, was not within the knowledge sphere of the sages of old. Being conscious of our contemporary 13.8-billion-year story that has shaped and is shaping our human adventure is truly awe inspiring and needs to be integrated within our 'wisdom traditions', if they are to be truly wise. Although science tentatively frames an evolving vast quantitative 'exterior' cosmos, our contemplative traditions *point* us towards a correlative qualitative 'interior', which in post-metaphysical terms (Murray 2019), can be experienced as expansive, spacious, wonder-inducing and transformative for our being, when their insights and practices are applied in our so called ordinary everyday lives. The overlaps identified between the respective contemplative paths of 'east and west' reveal a remarkable common cause during a period in which communications and cross referencing was not supported by the material architecture that we are presently familiar with. It was all communicated through an 'intranet', we might say.

Harvard psychologist and meditation teacher, Brown (1986) for example explored this topic of *universal deep structures* across the source texts of three major traditions, Hinduism, Theravāda and Mahayana Buddhism and derived his conclusions from a precise analysis of the technical language used in the Yogasutras, The Vissudhimagga and The Mahamudra traditions. It is described by Wilber (2006, p. 77) as "an absolutely brilliant and enduring classic in meditative stages". The yoga sutras of Patangali are thought to have been compiled c.a. 400 CE and comprise 196 sutras variously divided into chapters on Samadhi/absorption, Sadhana/practice, Vibhuti-siddhis/powers, and Kaivalya/moksha/liberation. The Vissudhimagga by Buddhaghosa (voice of the Buddha) is a comprehensive distillation, summary and analysis of the Tripitaka, the canons of Buddhist scriptures, Sri Lanka c.a. 430 CE. The 'Mahamudra' comprises a body of quintessential teachings from Tibetan Buddhism, the root text being the *Phyag chen zla ba'd od zer* of Tashi Namgyal or 'Moonbeams of Mahamudra', written in the 16th century.

Significantly, the claim is made by Brown that the distilled deep structures from his rigorous study are *universal* and operate across traditions. He states "[t]he models are sufficiently similar to suggest an *underlying common invariant sequence of stages*, despite vast cultural and linguistic differences as well as different styles of practice" (Wilber 2000a, p. 131, italics in original). They are in effect *trans-lineage*, while their 'surface' features differ, being variously expressed via the matrix of language, culture, social systems, self and biology, etc. Thus contrary to perennialists, who it is claimed fall prey to *the myth of the given* i.e., a lack of epistemic awareness in relation to the structuring of consciousness, where all paths are said to lead to the same enlightened end, Brown argues the evidence suggests "there is only one path but it has several outcomes, there are several kinds of enlightenment, although critically *all free awareness from psychological structure* [ego] *and alleviate suffering*" (Brown 1986, pp. 266–67, my italics) which is pertinent for our topic.

It can be noted also that these cartographies of liberation could analogously draw on Western models (Chirban 1986; Keating 2006, 2008), as found in the 'way of purification', the 'way of illumination' and the 'way of unification' (Underhill 2019), and indeed are included in Wilber's (2000a) integral psychological modelling. In the latter, around a third of one hundred plus comparative charts of human psychological development are contemplative systems, distilled from all the world's major wisdom traditions. Stanich's (2021) recent *Integral Christianity* for example also drinks heartily from the wells of the contemplative dimension of the Christian tradition.

It is noteworthy to highlight the significant post-post-modern or 'integral' nature of this claim, with its supposition of a universal underlying deep structural path. Although at first, this may sound similar to that long critiqued by postmodern scholars, the claim of access to universal deep structures purports to transcend, include and penetrate the postmodern critique of modern (surface) universals and essentialism. Distinct in both *surface* and *depth*, this universal claim may be appreciated within the epistemic and onto-logical assumptions of an *integral participative worldview* (as distinct from a modern and or

postmodern worldview), which seeks to integrate knowledge and wisdom across 'east and west' in a timely and timeless manner (Di Perna 2014, n.192).

So if we assume, for the time being, the veracity of this global phenomenon of 'waking up', as testified, recorded and cultivated within our contemplative traditions beyond an exclusive identification with 'ego', which can be understood as a necessary but transitional stage in our development, we enter the contextual birthing canal (epistemic and onto-logical), wherein 'unique personhood' can emerge. In order to elaborate on this case, I will employ insights from developmental psychology, a significant recent emergent in the scheme of things, in contrast to the millennia of the contemplative traditions just referenced.

### 3. Adult Developmental Awareness

Suzanne Cook-Greuter (2010) within her seminal publication *Post Autonomous Ego Development,* taking due cognizance of many other adult developmental theories, grounds her own empirical work in the further development of Loevinger and Blasi's (1976) Ego Development Theory (EDT). Her research is widely applied in a myriad of contexts, including in education, business and leadership programmes, with an analysis completed on "more than 9000 tests in more than 200 different academic and business contexts" (McNamara 2013, p. 209). A clear observation from her model is that the goal of Western socialisation for fully functioning adults is what she calls, for relatively transparent reasons, the conscientious/achiever self. Save the requirement of subtle detailed distinctions, mutatis mutandis, this 'structure-stage' of adult development correlates with many other veins of associated research in the field, (Baldwin [1906] 2000; Piaget and Inhelder 1958; Kohlberg 1981; Fowler 1981; Gilligan 1982; Gebser 1985; Kegan 1998, 2009; Commons et al. 1998; Beck and Cowan 2006; Fischer and Bidell 2006; O'Fallon 2012; Torbert 2021). In brief, this *dynamic structure-stage* (as distinct from a common static misperception of 'structure as form') of adult development, can be understood as indicative of the 'scientific rational mind', with all its brilliance and probing capacity, which has led many of us to enjoy 'the goods' of modernity and 'progress'. This structure-stage defines what it is to be an adult from a Western perspective. However, from a developmental psychology perspective, it relies on an unchallenged 'big assumption' of a subject and object distinction. As Cook-Greuter (2013, p. 19) elaborates:

> By most modern Western expectations, fully functional adults see and treat reality as something preexistent and external to themselves made up of permanent, well-defined objects that can be analyzed, investigated, and controlled for our benefit...Most adults . . . are not concerned with the basic arbitrariness of defining the objects. They are quite unaware that according to Koplowitz "the process of naming or measuring pulls that which is named out of reality, which itself is not nameable or measurable". They assume that subject and object are distinct, and that by analyzing the parts one can figure out the whole.

One potential takeaway from the above is the recognition that our 'everyday reality' is not an ontological given, independent of our perspective. Such a 'naïve' view potentially falls prey to 'the myth of the given'. Rather, the supposition is that we 'enact' reality according to our developmental awareness. Those familiar with integral metatheory may recognize the more complex formulation of a tetra-enaction of reality, according to our AQAL+ shadow constellation (Murray 2015), a distillation of significant vectors, veins and occlusions in human development, with their systemic implications (Bhaskar et al. 2016). This includes at least a recognition of the developmental shaping influence of/on our bio-psycho-socio-cultural conditioning/conditions. Our enacted reality includes not simply our contemplative state-stage development, and what I am referring to as our 'level of being', but significantly also one's structure-stage of consciousness development, or our 'level of becoming'. What merely *subsists* at prior levels *exists* at emerging levels, as they are 'objectified' and become conscious. In short, the nature of such development entails the subject of one stage becoming the object of the subject of the next, and until this developmental transition, we are deemed to be embedded in our views. As Kegan

(2009, p. 52) articulates, "the subject-object relationship, becomes increasingly expansive at successive levels of mental capacity." A positive point to highlight is that empirical evidence from the developmentalists listed above and emerging work from (Murray 2020; Shannon and Frischherz 2020), with considerable nuance, accounting for complexity, simplicity, shadow and without falling prey to simple "growth to goodness" modeling, (Stein 2010), suggests an evolutionary potential towards a desired increased capacity for perspective taking, 'metathinking' and a more 'coherent' experience of reality. What is evident is that contemplative awareness (level of being) needs also to be acutely conscious of developmental perspectives (level of becoming) and their implications for how we view the world and interpret our respective contemplative traditions and their 'original' teachings.

The linkage of change leaders' (or therapists) 'action logics' or developmental levels to organisational transformation and/or with nuance to therapeutic transformation, constitutes a significant 'developmental intuition' within the literature (Forman 2010). According to Torbert's (2021) extensive work in the field, "the change leaders' personal developmental action-logic [when at these later levels, predicted 59% of the variance of the success] is the single most important factor in whether an organisation transforms" (ibid, p. 443). Similarly Brown (2011, p. 1) in "an empirical study of sustainability leaders, who hold post conventional consciousness", documents "how leaders and change agents, with a highly developed meaning making system, design and engage in sustainability initiatives" and lists "access [to] non [post]-rational ways of knowing and [their] use [of] systems, complexity, and integral theories", as a principal finding of his empirical research. Aligned with the above, Scharmer (2018, p. 7), in distilling a core insight from his transformation research, with beautiful simplicity posits, that "the success of an intervention depends on the interior condition of the intervener". While much could be said regarding 'post rational' ways of knowing, it may be optimal for contemplative practices aiming at 'deconstructing ego', to consciously align with developmental perspectival awareness, given its enactive role in cultivating these evidently transformative and effective interventionist capacities. And towards building what O'Fallon (2010) terms "natural therapeutic holding environments" to catalyse 'post-egoic' and personal development.

## 4. Developmental Implications for Contemplative Paths

Now, within the context of Eastern contemplative-'Enlightenment' teachings, from which Van Gordon et al. (2016) principally draw, a 'traditional' path predominantly assumes an 'impersonal' realisation of identification with 'Source', where 'one' is an expression of this Authentic Self and this Authentic self is One 'Being'. However, the metaphysical and ontological assumptions of traditional paths have been called into question by further developments in modern and postmodern scholarship, originating with Kant's (2008) critiques and the consequent 'turn to the subject' in philosophy i.e., epistemology. The subsequent intersubjective turn along with a compelling revindication and differentiation of ontology, from its prior pervasive conflation with epistemology, i.e., the "epistemic fallacy" (Bhaskar 1975, 2008, 2012), has led many to recognise the subtleties of a "post-metaphysical" perspective in the social sciences, philosophy and spirituality (Habermas and Cronin 2017; Wilber 2006, 2017; Murray 2019), which seeks to articulate and affirm 'truths claims', keenly aware of their fallibilistic and provisional nature, from a humbler perspective. And it is problematic, from an ontological and developmental perspective, to claim as Van Gordon et al. (2016, p. 1) do, that our "imputed self" or ego is a "maladaptive condition whereby an individual is addicted to the belief that they inherently exist".

Much confusion in Western spiritual circles can derive from the pearls and perils of traditional interpretations of Buddhist *no-self* teachings, which when not understood in its appropriate context can have a debilitating impact on attempts to efface the ego, rather than embrace an integrative developmental dynamic (Wilber 2000b, pp. 717–34). On one hand, we can note the adage that we first need to develop a strong healthy ego in order to transcend *and include* 'it' (noting (Van Gordon et al. 2016) *emphasis on transcend* alone), or risk attendant mental health issues (Engler 1986). We paradoxically

note that the ego is implicated in the desire to eliminate the ego. Cook-Greuter (2013) cites Chogyam (2002) in *Cutting through Spiritual Materialism* as "perhaps the most cogent analysis of this mechanism," of how the ego is able to usurp ego transcendent moments or states, for its own vain glorification. Thus, while acutely acknowledging pervasive adult 'developmental issues' and the "impaired functionality" that the authors allude to, healthy 'ego development' can evidently be understood, au contraire to Van Gordon et al. as *adaptation* in action at a certain stage or stages in our psycho-spiritual growth (Cook-Greuter 2010). A matured awareness of 'ego states' can duly assist in catalysing ego-transcendence, with regard to integrating the 'individuating' and 'participative' functions of the psyche, which I will discuss further below. An array of contemporary spiritual authors and scholar-practitioners acknowledge, with nuance, that healthy ego development is a prerequisite stage(s), for balanced psycho-spiritual development, participatory enactment and is part of the gradual process of growing 'spiritual individuation', as cited in Ferrer (2017).

Perhaps more pertinently, from an ontological perspective, it is the case that *we always already assume we exist*, as to assume anything other than this involves a significant performative contradiction. This being the case insofar as our very actions reveal tacit assumptions, or a deeper belief, with which we may consciously theoretically disagree but nonetheless, can be excavated through our behaviour. As Bhaskar (2002, p. 70) suggests, "the source of the paradoxical nature of the self is as follows; whatever it is that is said about the self, there is something other than that which is tacitly presupposed", given that as Murray (2019) also reiterates, "all theories are underpinned by, usually tacit, ontological assumptions . . . which are deep, omnipresent and unavoidable". Therefore, if the ego (in the sense of EDT), is a necessary but insufficient stage(s) of our psycho-spiritual development and, as recognised by ego psychology, is 'a construct' (Loy 1999, 2002), what or who is it (without falling prey to transcendental reductionist assumptions), do we *inescapably* assume to 'be real', 'to exist' in ourselves and in each other?

## 5. Western Enlightenment

Conversely, in the Western Enlightenment tradition, one is alerted to an acute appreciation of the dignities of 'the individual', along with a recognition of the truly evolutionary and dialectical nature of our being and becoming. As Bhaskar (2002, p. 70) comments, "whatever the self is, it is clearly very important in contemporary society, being the bearer of legal, social, religious rights and responsibilities". This carries profound implications and is accompanied by the primary ethical injunction to treat 'each other' as an end in their selves, and to never instrumentalise each other as a means, echoing *the golden practice* or rule. This latter view is exemplified for instance in the societal outlawing of traditional slavery (as distinct from the ongoing systematic nature of 'modern slavery'), and the gradual enfranchisement and inclusion of respective 'out groups' and persons within society over the centuries and decades. This is in stark contrast to the simultaneous egregious dynamics of this enlightenment value of the individual being oppressed and 'one' being seen only as a means, to ideologically gratifying, warring and pervasive consumer ends (International Labour Office and United Nations Children's Fund 2021). In many respects we, at our peril, undervalue our 'personal rights' when failing to acknowledge their historical emergence as a significant recognition of not just a quantitative individual, but a qualitative affirmation and anticipation of our respective integrity of being.

And if *skillful means* connotes working within the context we find ourselves, an application of Buddhist psychology to western clients may do well to bear the bearer in mind, so to speak, while simultaneously recognising that the 'individualism' so valorised in western culture, is a mere shadow, a 'demi-reality' (Bhaskar et al. 2016) of what the western enlightenment's 'original insights' intimated (Taylor 1992). As the ethicist, Neil Levy (2018), maintains:

> With few exceptions, work on moral responsibility in the Anglophone world is resolutely individualist. The individual is not merely the primary unit of analysis and bearer of value; for the most part, individualism is taken for granted to such

an extent that philosophers are no more aware of their individualism than fish are of the water in which they swim.

In brief, the contention is that individualism, as a contemporary social phenomenon in large measure reflects a 'modern self', an ego identity, largely unaware of its own development, which through interdependent construction and shadow dynamics, plays out symptomatically and systemically in our historical moment. The social phenomena of hyper-individualism, narcissism, and relativism are a further intensification of these dynamics in our post-modern culture (Lasch 1979). We can, however, no longer ignore en-masse 'the bads' of modernity such as addiction rates, pollution, climate change, biodiversity loss and child labour for the production of many of our labelled consumer 'goods' and a host of other so called 'wicked problems', indicating symptoms of malaise, that have risen to variable levels of consciousness to characterise our epoch. According to the UN Human Development Report 2020, this age of the Anthropocene "means that we are the first people to live in an age defined by human choice, in which the dominant risk to our survival is ourselves".

And so, aligned with OAT's intent, Edwards (2016, p. 69), while expanding on this implication, remarks:

> Many of these predicaments are self-induced in that we believe in and utilise inadequate political, cultural, religious, scientific and commercial ideologies and their associated identities and practices to deal with these ills and consequently end up reproducing them in new and sometimes even more vicious forms. All this is taking a massive toll on the viability of the planets systems.

Problematic individualism is thus implicated in addiction and no less the Anthropocene. As ironic as it may initially seem, the remedy for individualism and our 'self-induced' predicaments, is a thoroughgoing individuation. Healthy individuation can be understood as an integration of contemplative awareness, where a more conscious union, communion with ones 'ground', 'being' and/or the whole is cultivated, beyond an exclusive identification with the ego, with an acute perspectival awareness, as reflected in the adult developmental literature cited above. It is an invitation for all, by virtue of our 'humanity', akin to that which the integral scholar Stein (2019, p. 274, italics in original) recounts, in that:

> the Judeo-Christian tradition contains a radical enlightenment teaching, with a message about the collective awakening of *everyone everywhere* . . . an Absolute Democracy in which each must live so that all will have the ability and dignity to be heard, known and counted. [And pertinently given our topic] . . . the core innovation enabling the democratisation of enlightenment . . . *is the reclaiming of the personal*".

## 6. Personal Perspectival Awareness

Thus, the contention emerges that the conflation of the 'personal' with the 'ego' is an unnecessary confusion, and potentially necessitates a catalytic re-orientation in ones 'self-system'. As Stein (ibid, p. 275) further notes, "the universal finds its expression only in and through an infinite variety of uniquely personal forms". This is significant insofar as our present spiritual landscapes are replete with traditional perspectival teachings conflating the personal and egoic, with all attendant risks for abuse and dehumanisation, when 'personal qualities' and an integrity of being are effaced and dissolved (ASI 2021). Notably Ken Wilber (Gafni 2012, p. 398, my italics) has recently awoken to this realisation in that even in non-dual states when:

> you are one with everything that is arising...you still feel a Unique Perspective on how this arises in your experience. [which] would *traditionally be interpreted* as an egoic holdover . . . [and] *prevent you from acting in the world on that uniqueness*. And all that really does is gum up action completely.

Wilber (ibid, my italics) thus posits:

> [T]he one true self is realising for the first time that it can manifest and embody in all these different perspectives and *not just force all of them to be reduced to the perspective of the One True Self'* . . . there's still just One True Self and it's the same I AMness arising in all these perspectives *that makes them real*.

A subtle recognition of the pitfalls of *transcendental reductionism* is evident here, whereby, as mentioned prior, the many are reduced to the One, and it is assumed, there are 'no real many'. In this awakening process Wilber thus discloses a pithy pointer, insofar as the realisation of *True self + Perspective = Unique Self.* He celebrates this individuation process in that, "[T]his formula . . . suddenly lights up all the individuality of the individual organism that had previously gotten wiped out [deconstructed] on the [traditional] way to discovering the One True Self" (ibid).

Gafni (2012) further emphasises the 'personal' nature of this 'unique self' insofar as he distinguishes levels of the personal, and makes the distinction between a conventional appreciation, level 1 personal, a egoic sense of self, and level 2 personal, which transcends a separate sense of self into communion, union with Self, whilst recognising that Self has "a personal face living in you, as you, through you" (ibid, p. xxvi) He describes the Unique Self not as a concept but:

> . . . a quality of presence when the ego is set aside, -even temporarily- . . . In Christianity this exercises itself as the personal relationship with Jesus Christ that so many sectors of Christendom understand so profoundly-and which is so derided by New Age teachers caught up in the impersonality of so much misunderstood Eastern teaching. In Hinduism the goddesses hold you in radical personal embrace. In classical Judaism the G-o-d loves you and knows your name while in Sufism and Kabbalah she is your most intimate erotic partner. None of this is dogma. It is the first and second person personal realisation that the interior of the face of the kosmos (sic) is the infinity of intimacy. And it is the revelation of the infinite, who yearns to love you personally and uniquely. (Gafni 2012, p. 109)

The supposition here is that this *personal perspectival awareness* was not well understood by traditional contemplative paths, and that it is a relatively recent 'recognition', rigorously researched and articulated, by adult developmental psychology, as referenced above. Personhood thus in the sense employed here, may be tentatively understood as this ontologically assumed, *pre-sense*, that brings *coherence, co-creativity, unity, value and meaning* to our many 'moving parts', in our interaction with our 'environment'. As humans we are compound interdependent beings, as testified by even a meagre appreciation of our present knowledge that 'normal matter', the 0.5% visible portion of our double dark or lambda $\Lambda$ CDM Universe, is constituent of our very physical bodies, which are ever growing and changing. We are creatures with 'will', 'emotions', 'feelings', 'brain', 'mind', 'memory', 'states', 'intellect', 'multiple intelligences', 'soul', and 'spirit'. All of these to variable degrees in relation to/with/in our surrounding society-systems-environment-nature-cosmos and with other living beings and humans. While we can readily testify to the experience of people (and systems) 'falling apart', 'breaking down', and all the attendant developmentally related physical, emotional, mental, spiritual and social health issues (not least addiction in all its guises), the extent to which we also cohere and unify these parts of our being and becoming is quite extraordinary. Indeed, it might be argued that this objective of cultivating *coherence* is at the very heart of our contemplative and psychological traditions.

*Personhood* may thus be understood as that unifying and mediating quality that integrates our parts into a meaningful whole, and it is from that pre-sense of 'wholeness' that we can truly and deeply connect and feel connected with others, 'systems' and nature, through a sense of continuity and identity in who we are, while many of our parts can change over time. Indeed, Bhaskar (2012, p. 168) claims that "unless [we] were in [non-dual, unitive] state[s] at least some of the time, you could not do or be anything at all". As he (ibid) points out:

Notice . . . in . . . your moment of reading, listening, hearing, following me, your moment of communion, not just your ego, . . . but your body dropped away . . . your cravings, the blind tenacity of your belief in the (exhaustive) physicality of being. And still you were you.

We recognise our family members for who they still are, while so many of the moving parts can and do change. People 'change their minds' regularly about matters and/or grow old as their bodies change. This may of course be extremely challenging as in the case of dementia, where there is a 'loss of self' and other parts of the person, when symptoms like amnesia, agnosia, aphasia and apraxia, etc., are present. However, research in the field, even in the case of advanced dementia, gleans qualitative reports of feeling "still the same" as before its onset, and for ensuring optimal care, strongly advocate that of "utmost importance is that other persons understand that persons with advanced dementia still are persons and support them to feel valuable" (Norberg 2019). Thus, the personal can be understood as relative individuation within the whole and is to be cultivated rather than conflated with the ego, as it implicitly is, in OAT's conventional interpretation.

**7. Radical Ontology**

John Heron's (1992) theory of personhood, with its extended epistemology, supports the ontological view being proposed here and proposes that core polarities of the psyche toward individuation or participation evolve somewhat predictably within patterned states, as a person matures and learns to balance and harmonise the two modes. He describes the ego as "that experience, [and/or 'state'] where we are out of balance, insofar as we are overidentified with the individuating mode at the expense of the participative". While a lack of space will not facilitate for an in-depth exploration of the implicating causal factors, it is interesting to note *the participative function* and the associated nuanced understanding of 'feeling', as distinct from 'emotion', is described by Heron (ibid, p.16) as:

the capacity of the psyche to participate in wider unities of being, to become at one with the differential content of a whole field of experience, to indwell what is present through attunement and resonance, and to know its own distinctness, while unified with the differentiated other. This is the domain of empathy, indwelling, participation, presence, resonance and such like.

Heron's (1998, p. xi), model thus critically presupposes:

. . . the human person [as] a distinct spiritual presence in and non-separable from the given cosmos and as such is not to be reduced to, or confused with, an illusory, separate, contracted and egoic self, with which personhood can become temporarily identified.

This echoes Teilhard De Chardin's (1959) profound contention that the artificial separation between humans and cosmos, lies at the root of our contemporary moral confusion. Jorge Ferrer (2017, p. 15) similarly emphasises the "key difference between modern individualism and spiritual individuation is thus the integration of radical relatedness in the later". The practical and ethical significance of such a 'radical ontology' of personhood cannot be overstated, insofar as 'we-space' research (Gunnlaugson and Brabant 2016) also signals how profoundly such an ontology impacts on collaborative success (or lack thereof) in groups, teams, organisations, communities and indeed, given our 'metacrises' (Rowson and Pascal 2021) one might add for, 'nations'. As McCallum et al. (2016) states, with relevance to an ethic of care:

For us as practitioners, this philosophical view of differentiated and yet unified field of consciousness provides a way of understanding the radically interdependent nature of relationships between parts and whole, individuals and groups, and sub groups within larger and larger collectives . . . in the instance of individuals who are recognised with genetic or social "frailties", the degree to which a community understands these individuals not as "other", but as being

integral parts of a larger whole will determine approaches to care, allocation and resources, etc.

The distinction of the 'person', with a core "capacity for feeling" as a spiritual presence and the 'ego', as an alienated part of the psyche "over identified with the individuating mode at the expense of the participative", is understood as an emergent experience within Heron's "states of personhood" (Heron 1992, p. 53). Heron's work strongly correlates with other literature and research within 'the field' (Merry 2020), a selection of which have been cited above. Heron too takes substantive ontological issue with the seeming, hegemonic transcendental reductionism of some traditional eastern teachings, as in the OAT's interpretation, and their implications i.e., positing that "the many are reduced to the one via the concept of illusion: there are no Real many, only the unreal many, illusionary selves that ultimately disappear in the light of the One" (ibid, p. 10). Heron (1998, p. 80, italics in original) critiques this view as "a *repeal* of a conservative creation model," as distinct from cognising the profound import of an evolutionary worldview, for all the contemplative traditions, as exemplified in the momentous, but still relatively unknown 'Fourth Turning' in Buddhism, as we shall observe shortly. Interestingly Aurobindo (2005, p. 482), while echoing the premise in OAT, "[t]he one thing that can be described as an unreal reality is our individual sense of separativeness (sic) and the conception of the finite as a self-existent object in the Infinite,"also critically asserts:

> [t]he true Person is not an isolated entity, his individuality is universal; for he individualises the universe: it is at the same time divinely emergent in a spiritual air of transcendental infinity, like a high cloud-surpassing summit; for he individualises the divine Transcendence, (ibid, p. 1008).

Heron (1998, p. 79) is thus concerned that "in elevating the human to the absolute, it ignores the asymmetrical relation between the finite and the infinite" ..and regards it in his model, as "an illusionary state of spiritual inflation". He (ibid, p. 82) thus maintains that "it is important to challenge these claims for the very good reason that they can, for a while at any rate, intimidate and disempower some people from making deep, creative choices about their own spiritual path".

It is also noteworthy is this context that while Van Gordon et al. (2016) qualify a psychopathologising of "belief in god," the assumption from "the Buddhist perspective" that "a belief in a divine and/or ruling being requires that there is a self," is problematic in conception and practice. It is also somewhat ironic in the context of the very aims of OAT, as it is precisely an integrative appreciation of '2nd person concepts of Spirit', or an 'i-thou' 'devotional' practice (central also in Tibetan Buddhism) which potentially recognises higher-deeper, broader, transcendent, immanent and situational levels of thou, "that before which the ego is humbled," (Wilber 2006, p. 160) and which facilitates the cultivation of the 'other' centered favourable character traits, espoused in the OAT approach. As Wilber (2006, ibid) expresses, "[i]n short failing to acknowledge your own Spirit in 2nd-person is a repression of a dimension of your being-in-the world". While we can no doubt acknowledge a pervasive developmental conflation of 'God' with traditional mythic perspectives alone, a developmental orientation which appreciates the '123 of God' (ibid, p. 161), or first person, second person and third person approaches, integrally recognising perspectives and depth, 'East and West' is *more becoming* in a conscious evolutionary age (Corless and Knitter 1990).

Stein (2019, p. 279) likewise notes "the widespread failure to understand this radical truth about the [democratic and personal] nature of enlightenment has kept it from being a legitimate modern belief and aspiration". In a similar vein, De Chardin (1959, p. 283) makes an explicit cultural connection between the discovery of "the sidereal world, so vast" and what he refers to as the *depersonalisation* or *impersonalisation* of modern man, echoing more contemporary insights and remedies from the cosmology of Abrams and Primack (2011) where instead of 'modern humans' feeling lost and insignificant in the vast

cosmos, we appreciate our profound integrality with the whole. De Chardin (ibid, p. 285) maintained:

> [F]ar from being mutually exclusive, the Universal and Personal (that is to say 'centred') grow in the same direction and culminate simultaneously in each other. It is therefore a mistake to look for the extension of our being or of the noosphere in the Impersonal.

Fittingly, de Chardin poses the question, "what is the work or works of man if not to establish, in and by each one of us, an absolute original centre in which the Universe reflects itself in a unique and imitable way?" (ibid, p. 287). This characteristic of evolution, he claims, is underscored in any domain, "whether it be the cells of the body, the members of a society or the elements of a spiritual synthesis", (ibid, p. 288) in the principle *union differentiates*. Teilhard thus maintained:

> [T]he peak of ourselves, the acme of our originality, is not our individuality [ego as over identified with the individuating function] but our person; and according to the evolutionary structure of the world, we can only find our person by uniting together...[though] not every kind of union will do . . . it is centre to centre that must make contact and *not otherwise* . . . [as] the true ego grows in inverse proportion to 'egoism' (ibid, pp. 289, 290).

## 8. Evolutionary Awareness-A Fourth Turning of Buddhism

The generative implications of our present developing awareness that we live in a vast evolutionary universe is shaping what many are now referring to as the 'Fourth Turning' in Buddhism (Wilber 2014) and somewhat contrary to that which Van Gordon et al. (2016) suggest, regarding interpretations of prior turnings, may well amount to significantly more than another "variation on the same theme" (ibid, p. 4). It may indeed hold significant import for interpretations of "no-self" teachings, the "innermost aspect of consciousness", "transmigration" and "*pashchimadharma* [Sanskrit] or *mappō* [Japanese]" teachings, not discounting their relative insights, within the consciousness of a vast evolving and cornucopian universe. Indeed each developmental structural stage of 'spiritual intelligence' governs how persons interpret their contemplative/spiritual experience and/or their respective traditions, with a recognition that "the very core of the enlightenment experience will change from stage to stage," (Wilber 2017, p. 9) as we enter anew the hermeneutic circle, moment to moment (Panikkar 1979).

As the reader may recall, in brief, the First major Turning of the Buddhist wheel of dharma refers to the teachings of Siddharta Guatama, represented by Theravāda Buddhism. The Second Turning refers to Nāgārjuna's teaching on 'emptiness', sūnyatā, within the Madhyamika and foundational for the Mahāyāna and Vajrayana schools. The Third Turning focuses its teachings on 'Buddha nature', or *tathāgatagarbha*, embryonic Buddhahood, implying 'enlightenment' is our true and natural state of mind and is represented by the Yogācāra school. This contemporary Fourth Turning of the Buddhist wheel of dharma, (or variously Fifth, if counting tantric/esoteric Buddhism) includes our own era's ongoing discovery of 'evolutionary theory'. It therefore recognizes that the very world of 'form', is itself evolving and if as previously taught, 'emptiness and form' are not two, i.e., nondual, the Fourth Turning in essence recognises that 'emptiness and evolving form', are not two, i.e., nondual. This same complexification of form (for e.g., from strings, quarks, atoms, molecules, to cells, to multicellular organisms, etc.,) is also occuring in humans as attested by the literature referenced, indicated not least by our increasing developmental capacity for perspectival awareness. Thus the supposition is that while tradtional enlightenment remains unchanged in its *Freedom* (Emptiness) aspect, its *Fullness* (Form) has evidently continued to evolve. And as the formula of true self plus perspective indicated, the realised 'true self' is now being expressed in this Fourth Turning through what Wilber refers to as "a post egoic nondual realisation of unique perspective", (Gafni 2012, p. xx) with its attendant ontological, personal and ethical import, as depicted above.

## 9. Conclusions

Van Gorden et al.'s Ontological Addiction theory presently understood as "the maladaptive condition whereby an individual is addicted to the belief that they inherently exist" risks being enmeshed in a performative contradiction. This is related to an implicit transcendetal reductionsist assumption that is operative in its conception. Any assimilation and an application through skillful means to mental health within a western context will also seek to integrate the insights of the Western Enlightenment and the value of the individual. Critically, this entails a developmental appreciation of the problematic perception of egoic individualism as distinct from the conception of an individuating 'whole person', with ontological import. Thus OAT could positively be supplemented, reconstructed and reconceived as an Ontological Affirmation Theory.

Van Gordon et al. (2016) are correct when they indicate that present mainstream western bio-psycho-social models of medical and scientific opinion are inadequate and operate somewhat broader but shallower in comparison to Buddhist psychology, in their core assumptions of the determinants of psychopathology. This, however, chiefly owes to the lack of a depth ontology in mainstream western models and the incompletion of their own enlightenment project, which critially needs to source its contemplative wellsprings for sustenance, given a present pervasive pyschopathology (in the imparied functional rather than statistical sense) of individualism.

However in contradistinction to Van Gordon et al. (ibid) 'the ego' as perceived through a thoroughgoing developmental lens can be understood as already potentially 'adaptive', insofar as the literature indicates, 'it' evidentially evolves and potentially matures, into a full and flourishing integrative (individuating and participative) personhood. This is aligned with Van Gordon et al.'s (2016) "dynamic and non-dual self", but is critically distinctive, insofar as it affirms and explicitates the radical ontological and ethical implications.

A potential course corrective for OAT, from its implict trancendental reductionism is applicable, if the profound implications of an evolutionary worldview are integrated within the contemplative tradition it draws upon. This is pointedly exemplified and foreshadowed by the momentous Fourth Turning in Buddhism, along with more recent findings in adult developmental psychology, providing possible diagnostic markers in relation to the "therapeutic (and spiritual) discernment [that] is clearly required in order to assess the suitability of a particular individual to receive, and progress through, the various (generic) treatment phases outlined in [their] paper," (ibid, p. 27).

While aligned with Van Gordon et al. (2016) in recognising the core sense of lack that invariably accompanies an egoic sense of self, and its resultant endless cravings for external fullfillment, such as 'addictions' in a myriad of forms, the excavation of inescapable ontological presuppositions, as outlined, reveals who we are, as *persons,* can act as a potential course corrective, to avoid the possible pitfalls of transcendental reductionsim and the oft accompanying egoic inflations and deflations, when the pearls of 'no-self' become perilous. It also underscores a real 'integrity of our being' as we progressively realise the profound evolutionary and participatory insight that 'the whole universe' is, not least evidently in 'the material sense', already in us and we in the whole. Or as Van Gordon et al. (2016, p. 24) somewhat ironically relay it, "[a] person who has realized true self cares for the individual because they care for the whole, and vice versa". Would that they also make explicit the associated ontological presuppositions apt to cultivate and sustain this ethic of care, especially for those who inescapably assume 'themselves' to exist and *be real*.

**Funding:** This research received no external funding.

**Conflicts of Interest:** The author declares no conflict of interest.

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
