# Peer review of "The Significance of ‘the Person’ in Addiction"

_religions, doi:10.3390/rel12100893_

Round 1

Reviewer 1 Report

The content of this paper was quite interesting and thought provoking. I appreciated the detailed background and attention given to the different ways that self, person, and individual have been understood across various disciplinary perspectives. Tying those understandings more directly into how the “person” is conceived within the OAT modality would have helped make the connections clearer with respect to addiction treatment.

I was able to see the clear demarcations made between individualism and individuation, as well as the preference for a conceptualization of the interconnected nature of the self to that which is not the self. This was done quite well early in the paper, but as the paper continued there was some loss of connection to the critique of the OAT approach. By the conclusion section I was not able to see the rationale for re-naming OAT  Ontological Affirmation Theory; that could use a bit more unpacking.

As far as readability, I believe some editing and proof-reading would help. It was sometimes difficult to catch the meaning of particularly long passages that were sprinkled with quotations that were not clearly marked according to APA style (or another style for citing sources). It should be made more clear where the author’s narrative is and where quotes are being used to support author’s narrative. It will help to have double quotation marks used for direct quotes with citations nearby using page numbers. For block quotes it would help to indent (if more than 40 words) being sure the author/date/page number is clear.

It may be that I am simply not familiar with some of word choices, but creating shorter, more to the point sentences will help those who, like me, sensed there was some very good content within the complex wording. Sometimes the flow of narrative was cumbersome, albeit with careful re-reading the points made were very compelling! 

Author Response

Reviewer Comments: The content of this paper was quite interesting and thought provoking. I appreciated the detailed background and attention given to the different ways that self, person, and individual have been understood across various disciplinary perspectives. Tying those understandings more directly into how the “person” is conceived within the OAT modality would have helped make the connections clearer with respect to addiction treatment.

Response: As laid out in the reviewed paper, Van Gordon et al. 2016 conceive of the person as ‘ego’. They in the main make an implicit conventional level assumed conflation. I have amended the review article abstract to reflect this, for those who may not have had the opportunity to read the Van Gordon et al paper

Reviewer Comments: I was able to see the clear demarcations made between individualism and individuation, as well as the preference for a conceptualization of the interconnected nature of the self to that which is not the self. This was done quite well early in the paper, but as the paper continued there was some loss of connection to the critique of the OAT approach.

By the conclusion section I was not able to see the rationale for re-naming OAT Ontological Affirmation Theory; that could use a bit more unpacking.

Response: In short, the critique is based on an excavation of the assumption of a conflation between ‘person’ and ‘ego’ and presenting it as ‘a gap’ in the authors self-system. It seeks to expose a hidden performative contradiction, insofar as the authors themselves inescapably hold implicit ontological assumptions. And it excavates the implicit transcendental reductionist assumption, where the authors assume ‘the many’ are reduced to ‘the one’ and there are, it is assumed, no real many.

I argue that the many are real, and that each of us, has ontological import by virtue of our ‘personhood’, which is emerging and refracted through the western enlightenment tradition and with nuance through developmental psychology. An insight of ‘individuation’ is also emerging and recognised within an evolutionary conscious Fourth Turning in Buddhism and thus provides a potential rich resource, while simultaneously holding significant implications for Buddhist Psychology and its applications, particularly in the West. I contend that there are ethical consequences for an underdeveloped ontology of the person, as is presently the case with OAT, insofar as it has and can lead to ‘a failure to act’, in its conflation and confusion of person and ego. And alternatively propose that there are timely and significant developmental fruits (for e.g., an ethic of care) in affirming the ontological value of personhood, hence the potential reframe of OAT into Ontological Affirmation Theory.

.

As far as readability, I believe some editing and proof-reading would help. It was sometimes difficult to catch the meaning of particularly long passages that were sprinkled with quotations that were not clearly marked according to APA style (or another style for citing sources). It should be made more clear where the author’s narrative is and where quotes are being used to support author’s narrative. It will help to have double quotation marks used for direct quotes with citations nearby using page numbers. For block quotes it would help to indent (if more than 40 words) being sure the author/date/page number is clear.

It may be that I am simply not familiar with some of word choices, but creating shorter, more to the point sentences will help those who, like me, sensed there was some very good content within the complex wording. Sometimes the flow of narrative was cumbersome, albeit with careful re-reading the points made were very compelling! 

Editing and proofreading has been carried out, shortening some passages, which hopefully assist with flow, plus a style of “ ” has been employed.

All minor changes in the body of the work can be seen in green in the revised manuscript.

Personal note: I would like to thank the reviewers for their kind comments, constructive queries and suggestions, which I believe has further honed and elevated the quality of the review article.

Reviewer 2 Report

This article discussed Ontological Addiction Theory (OAT) from Van Gordon et al (2016). The author argued with presenting an alternative view on the ontology of ‘the person’. As distinct from a contracted separate egoic self, which the author proposed as significant for the ‘fuller’ experience of a reconstructed ‘dynamic and non-dual self’, as cultivated in their treatment.

This manuscript well demonstrated the possible argument on OAT on an alternative view on the ontology of ‘the person’. I think it is thoughtful and extended the consideration of OAT.

However, the part of “A Fourth Turning of Buddhism” may think carefully about the original theory of Buddhism. The “fourth turning” may be not the image about what the Christian world thinks.

Please provide more theories about ‘the person in OAT’ related to ‘the fourth turning’.

Author Response

Reviewer Comments: This article discussed Ontological Addiction Theory (OAT) from Van Gordon et al (2016). The author argued with presenting an alternative view on the ontology of ‘the person’. As distinct from a contracted separate egoic self, which the author proposed as significant for the ‘fuller’ (Response: updated with embodied) experience of a reconstructed ‘dynamic and non-dual self’, as cultivated in their treatment.

This manuscript well demonstrated the possible argument on OAT on an alternative view on the ontology of ‘the person’. I think it is thoughtful and extended the consideration of OAT.

However, the part of “A Fourth Turning of Buddhism” may think carefully about the original theory of Buddhism. The “fourth turning” may be not the image about what the Christian world thinks.

Response: The Fourth Turning stems from an appreciation of the significance of ‘evolutionary theory’ for all schools of Buddhism and is acutely cognisant of the hermeneutic implications in relation to interpretations of ‘original Buddhism’. The Fourth Turning stems from within the Buddhist fold as it were and fosters new potential within the field of inter and intra tradition/faith dialogue, especially in relation to ‘individuation’, as developmentally understood within respective traditions. This view is emerging as a realisation within an evolutionary conscious Fourth Turning of Buddhism and resonates a certain ‘homeomorphic equivalence’ with experiences of the person, as evolving and refracted through the Western Enlightement tradition and posits a fertile site for future further research.

Reviewer Comment: Please provide more theories about ‘the person in OAT’ related to ‘the fourth turning’.

Response: In short, a theory of ‘the person’ is not developed in OAT and is conflated with ego. I critique this view of Van Gordon et al. (2016) and present as distinct the developmental view of the person, which with nuance, is expressed through the Fourth Turning as “a post egoic nondual realisation of unique perspective”. As above, this view is emerging as a realisation within an evolutionary conscious Fourth Turning of Buddhism. Holding an ‘individuated quality of personal presence’, it carries significant ontologial and ethical import, not least for ‘Buddhist Psychology’ and its ‘self-system’, particularly as it is practiced in the West. My reframe of OAT as ontological affirmation theory postulates a more complex and coherent self-system, and with subtlety is grounded in empirical evidence from developmental psychology and integrates the subtle and causal state-stages of contemplative development. A fuller exposition of this ontologially affirmative stance and its implications in context is anticipated, where space will permit in a subsequent paper.

Personal note: I would like to thank the reviewers for their kind comments, constructive queries and suggestions, which I believe has further honed and elevated the quality of the review article.
